# Real-Time Location System (RTLS) Based on the Bluetooth Technology for Internal Logistics

**Augustyn Lorenc** [1,*] 📷, **Jakub Szarata** [2] **and Michał Czuba** [2]

1 Department of Rail Vehicles and Transport, Cracow University of Technology Poland, al. Jana Pawla II 37, 31-864 Cracow, Poland
2 SKK S.A., R&D, ul. Gromadzka 54A, 30-719 Cracow, Poland
* Correspondence: a.lorenc@pk.edu.pl; Tel.: +48-123743659

**Abstract:** The problem of object localization in indoor environments is very important in order to make a company effective and to detect disruption in the logistics system in real-time. Present research investigates how the IoT (Internet of Things) location system based on Bluetooth can be implemented for this solution. The location based on the Bluetooth is hard to predict. Radio wave interference in this frequency is affected by other devices, steel, vessels containing water, and more. However, proper data processing and signal stabilization can increase the accuracy of the location. To be sure that the location system based on the BT (Bluetooth) can be implemented for real cases, an analysis of signal strength amplitude and disruption was made. The paper presents R&D (Research and Development) works with a practical test in real cases. The signal strength fluctuation for the receiver is between 7 and 10 dBm for ESP32 device and between 13 and 14 dBm for Raspberry. For commercial implementation the number of devices scanned in the time window is also important. For Raspberry, the optimal time window is 5 s; in this time six transmitters can be detected. ESP32 has a problem with detecting devices in a short time, as just two transmitters can be detected in 4–8 s time window. Localisation precision depends on the distance between transmitter and receiver, and the angle from the axis of the directional antenna. For the distance of 10 m the measurement error is 1.2–6.1 m, whilst for the distance of 40 m the measurement error is 4.9 to 24.6 m. Using a Kalman filter can reduce the localization error to 1.5 m.

**Keywords:** real-time location system; Bluetooth; Bluetooth Low Energy; sensor; internal logistics; disruption; location





## 1. Introduction

The location of objects inside buildings is becoming more and more important for companies wishing to increase their efficiency and reduce their resource use. The use of GPS (Global Positioning System)/Glonass/Beidou/Galileo technology in buildings is not possible due to wave attenuation.

The currently available Real-Time Location Systems (RTLS) solutions are based on a location based on Wi-Fi, RFID (Radio-Frequency Identification) or UWB (Ultra-Wideband) technology. The comparison of this technologies are presented in Table 1. These types of systems enable the location and identification of objects in real-time in a closed and open environment, but due to the technology used, they cannot be used in a dynamically changing environment characteristic of large production companies. In such an environment, the accuracy of the location differs significantly from the actual location. It is influenced by the phenomenon of the refraction of waves, overlapping, and other types of disturbances caused by the diversity of the environment surrounding the locating transmitters [1–4].

**Table 1.** Comparison of object location technologies.

|  | Wi-Fi | Radio-Frequency Identification (RFID) | Ultra-Wide Band (UWB) | Bluetooth Low Energy (BLE) |
|---|---|---|---|---|
| Frequency | 2400 ÷ 2485 MHz<br>4915 ÷ 5825 MHz | 125 kHz up to 0.5 m<br>13.56 MHz up to 3 m<br>868 ÷ 956 MHz up to 6 m<br>2.4 GHz 5.8 GH up to 6 m | 3.1 ÷ 10.6 GHz in USA<br>6.0 ÷ 8.5 GHz in Europe | 2402 ÷ 2480 MHz<br>40 Channels (2 MHz Spacing) |
| Transmission speed | 900 Mb/s | 256 Kb/s | 2 Gb/s up to 10 m | 0.27 ÷ 1.37 Mbit/s |
| Standard | IEEE 802.11 | Tiris, Unique, Q5, Hitag, Mifare, Icode | IEEE 802.15.3 | Bluetooth 5.2 |
| Range | 500 m | 6 m | 10 m | 100 m—typical, 400 m maximum |
| Date of creation | 1991 | 1983 | 2002 | 2001 |

The difficulty in adapting the currently functioning solutions results from, among other causes, the need to build a basket infrastructure (power supply network, Lan/Wi-Fi network). Standard BLE (Bluetooth Low Energy) receivers can download data packets every 10 s. In addition, one receiver can only read one data packet for each transmitter in the same time interval. Therefore, it would be possible to read one packet at the beginning of the time interval and the other at its end. Bluetooth Low Energy (BLE) is a wireless technology designed and marketed by the Bluetooth Special Interest Group (Bluetooth SIG). BLE is designed for devices with low power consumption that operates in the 2.4 GHz ISM (The Industrial, Scientific and Medical) band. The goal of this technology is to connect devices over relatively short ranges. It is currently a very popular protocol widely used in IoT (Internet of Things), Industry 4.0 and Smart Homes. In addition, BLE technology support localisation solution such as AoA (Angle of Arrival), AoD (Angle of departure). Thus, the maximum difference between packets can be almost 20 s. This is a significant problem because the maximum speed of vehicles on the premises of enterprises can be up to 40 km/h. Therefore, the trolley could cover a distance of 111 m in 10 s. However, in the case of a human moving on foot, this distance would be about 14 m (not considering disturbances related to signal level fluctuations). Therefore, such a solution would be completely unacceptable. To ensure the accuracy in the location of objects, it is necessary to obtain a reading frequency of less than 1 s.

The main advantages of BLE are: very low power consumption; security-encryption using 128-bit AES (Advanced Encryption Standard) algorithms; it is cheap to implement; it can carry up to 255 bytes per package (message capacity); and frequency hopping partially eliminates interference from other networks. Its main disadvantages are lower data rates compared to Wifi or GSM (Global System for Mobile Communications) technology and that it cannot be used for long distance and is vulnerable to interception.

In the paper the unique research was made to analyse the disruptions and problems for using BLE technology for RTLS. This research includes:

-   Analysing signal strengths base on the channel,
-   Analysing correlation between environment and transmitter speed,
-   Analysing localization accuracy base on the sampling rate,
-   Analysing correlation between signal strength and transmitter speed.

Additionally, the developing work was made for the production of electronic devices, transmitter and receiver.

The article begins with describing the state of the art. Then the materials and methods are described. In this section the receiver and transmitter are described with object location methods (signal strength RSSI, Angle of Arrival, Time of Flight). The next section is the research and results chapter, which describes the sampling rate required to achieve the specified level of localization accuracy, the correlation of other moving objects with the

speed of the analysed objects and Kalman filter, and its influence on distance estimation. The article ends with a conclusion.

## 2. State of the Art

Many RTLS are used in the hospitals and clinics [5–7]. Lean methodology helps maximize value by reducing waste, first by defining what value and waste are in a system. In ophthalmology clinics, value is determined by the number of patients flowing through the clinic for a given time. We aimed to increase value using a lean-methodology guided policy change, then assessed its impact on clinic flow using an automated radiofrequency identification (RFID) based real-time locating system [8]. Similar systems are using for tracking the movements of the clinics' workers and increase the effectiveness of internal logistics [9]. During the period of the SARS-CoV-2 pandemic, RTLS were also used for reducing the transmission of virus in the hospital environment [10]. The indoor environment in typical building like this is quite easy to analyze. In typical solutions in different rooms the receivers are placed and scan the sensors (transmitters) in the range of the antennae. The antennae can be omnidirectional or directional. This depends on the exact place. For corridors, the directional antennae are better; in this case the RSSI can be used to predict the distance of the sensor from receiver. For rooms, the better option are omnidirectional antennae.

Due to No Line-of-Sight (NLoS), low signal strength, and low accuracy, GPS is not suitable to be used indoors. As a consequence, the indoor environment necessitates a different Indoor Positioning System (IPS). Different technologies, algorithms, and techniques have been proposed in IPS to determine the position and accuracy of the system [11]. Real-Time Location System (RTLS) has been confirmed to have a high suitability for various construction site applications in the past decade: resources tracking; productivity monitoring; labour and equipment safety; and robotics navigation [12,13]. Many similar solutions are uses for navigation systems based on the sensor networks [14]. Nowadays, accurate localization plays an essential role in many fields, such as target tracking and path planning [15]. The challenges of indoor localization include inadequate localization accuracy, unreasonable anchor deployment in complex scenarios, lack of stability, and high cost. To overcome these shortcomings, a comprehensive ultra-wide band (UWB)-based real-time localization system (RTLS) can be used [15]. The main advantages of UWB are low cost of infrastructure and high range of devices [16], [17]. The location data achieved by these frameworks can be used to enhance safety measures, improve the construction site layout management, as well as extensive efficiency analysis [18]. For stabilization the signal strength for precise positioning, a few methods can be used, such as the Signal propagation model, Trilateration, Modification coefficient, and Kalman filter [19,20]. Additionally, some IPS use different technology like RFID. RFID technology is used to track the location of people or things in an interior setting in real-time [21]. However, in this solution the cost of infrastructure is high and the range of devices is poor.

Location systems are becoming increasingly important in the implementation of Industry 4.0 approaches [22]. Location systems are becoming increasingly important in the implementation of Industry 4.0 approaches. RTLS for asset tracking provides useful data inputs for various purposes such as monitoring of personnel on the factory floor, equipment utilization, inventory levels and flow of work-in-progress (WIP). This in turn leads to better identification of bottlenecks, optimization of factory layouts, more informed planning and scheduling, and hence an overall improvement in productivity [23]. Furthermore, a similar system is used for value stream mapping. For example [24]. a semi-automated VSM solution was used that leverages dynamic Real-Time Location System (RTLS) data to track motion throughout a production floor. This solution incorporates a high precision UWB RTLS, which allows for the real-time analysis and formulation of different state maps to track processes, equipment, people, and materials on the manufacturing floor. Some researchers also use 5G technology [25] or 6G technology [26] for location systems. But nowadays, the main problem with this technology is poor 5G network operation areas.

## 3. Materials and Methods

### 3.1. Research Methodology

To analyse the possibility of using Bluetooth technology for developing Real-Time Location System, the analysis of signal strength in different cases was made. The authors prepared their own devices with both transmitter and receiver, and based on these, we analysed the signal for different environments, transition channels, static and moving objects, and different sampling rates (advertisement). A simple schema showing the research methodology is presented below.

### 3.2. Object Location Methods

Object localization is performed based on a set of receivers and transmitters using one of three methods:

- distance analysis based on signal level (RSSI)-trilateration method,
- signal reception account analysis (Angle of Arrival, AoA),
- time analysis between signal transmission and its reception (Time of Flight, ToF).

### 3.2.1. Method I—Signal Strength (RSSI)

The first of the described methods is based on the level of the RSSI signal (Received Signal Strength Indication), i.e., the strength indicator of the received radio signal. It should be remembered that it is not the same as the quality indicator. The value of the parameter is given in dBm, and its range is from 0 to 100, where 0 is a number representing the strongest signal level, corresponding to the shortest distance between the transmitter and receiver. Typically, the RSSI signal level is presented in negative notation.

The method based on RSSI signals is not very precise, but it allows one to calculate the potential zone location with the use of one receiver and transmitter. Formula (1) is used to convert the signal strength into a distance. The actual reading of the signal level at the distance $d$ is taken as the *RSSI* value. On the other hand, the value of *A* is taken as the signal level at a distance of 1 m from the transmitter, which allows the calculation of the $n$ coefficient. The parameter $n$ is therefore a signal calibration coefficient in a specific configuration (selected receiving and transmitting devices, antennae, working environment).

$$d = 10^{-\frac{(RSSI-A)}{10n}} \tag{1}$$

where the value of the $n$ coefficient is calculated from the formula:

$$n = -\frac{RSSI - A}{10 \log_{10}(d)} \tag{2}$$

where:

$n$—correction factor depending on the environment [db/m],
*A*—for the types of transmitters used, the RSSI measurement value from a distance of 1 m [db],
d—calculated distance between the directional antenna and transmitters, [m].

Calculating distances from three points allows one to determine the position of the transmitter in relation to the receivers using the trilateration method. Figure 1 shows a diagram of the trilateration process.

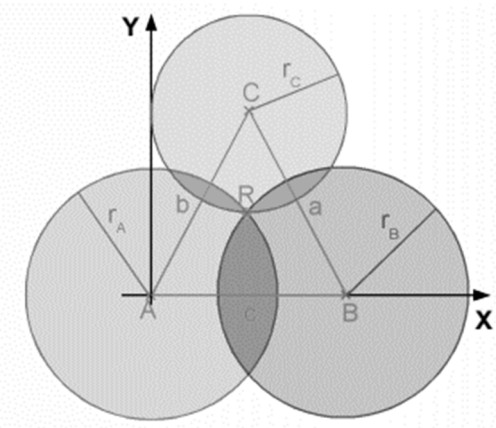

**Figure 1.** The method of determining the location of the transmitter by means of trilateration.

In the localization process, three signal receivers (A, B and C) are used to determine the location of a mobile object with a mounted transmitter (marked as R), which must be within the range of all three receivers. The positions of the receivers are known (defined at the installation stage), and the distances between them (a, b and c) are constant values. The variable distances between the object and the receivers are the radii of the circles $r_A$, $r_B$ and $r_C$, respectively. The radii of the circles correspond to the strength of the RSSI signal as measured by the individual receivers. As the location takes place in two-dimensional space (the height difference of the receivers and transmitters is 1–2 m, so the height has a slight impact on the determined distance), the circles are in fact circles on the $z = 0$ plane. Using three receivers is necessary to obtain an unambiguous result for positioning in two-dimensional space. Accordingly, the formulas defining the circles can be written as:

$$x^2 + y^2 + z^2 = r_A^2 \tag{3}$$

$$(x - x_B)^2 + y^2 + z^2 = r_B^2 \tag{4}$$

$$(x - x_C)^2 + (y - y_C)^2 + z^2 = r_C^2 \tag{5}$$

Assuming that the above Equations (3)–(5) form a system of equations, its solution to the variables $x$, $y$ and $z$ is:

$$x_R = \frac{x_B^2 + r_A^2 - r_B^2}{2x_B} \tag{6}$$

$$y_R = \frac{y_C^2 + x_C^2 - 2x_C x_R + r_A^2 - r_C^2}{2y_C} \tag{7}$$

$$z_R = \sqrt{r_A^2 - y_R^2 - x_R^2} \tag{8}$$

### 3.2.2. Method II—Angle of Arrival (AoA)

Angle of Arrival (AoA) is a technique that allows you to determine the direction from which a BLE packet came. This is the basis for triangulation, which is one of the methods of finding a location. The technique mentioned requires the use of at least two receivers (in the 2D model scenario). Having the angle of receiving a packet from the transmitter, we can determine the point of intersection by the receiving devices, which is the calculated location. The case is illustrated in the Figure 2 below.

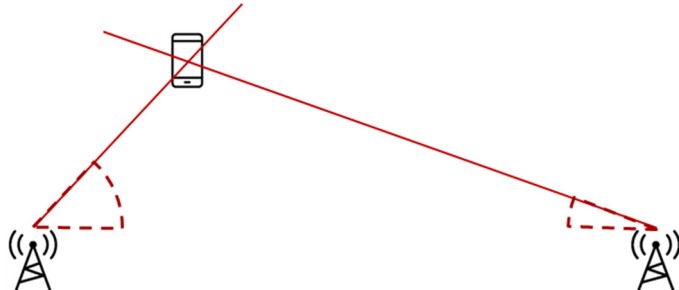

**Figure 2.** The method of determining the location of the transmitter by means of trilateration.

In order to obtain the angle in the receiving devices, it is necessary to use a minimum of three antennae that are within the transmitter's range at the same time. During scanning, it quickly switches between the given antennae. This makes it possible to observe phase shifts resulting from small differences in the length of the signal path to different antennae. The differences depend on the direction of arrival of the BLE packet, which in the AoA solution must contain a Continuous Tone (CT) section, where there is no phase shift resulting from the signal modulation.

### 3.2.3. Method III—Time of Flight (ToF)

Time of Flight (ToF) is a technique that allows the distance of the receiver to the transmitter to be determined based on the delay time travelled by BLE packets during data exchange between devices. Knowing the distance between the receivers and the transmitter, one can perform the triangulation algorithm. The method consists in drawing circles around receivers with a radius equal to the specified distance, in this case ToF. The intersection point represents the probable location of the transmitter being tracked. A typical requirement is a minimum of three receivers to find one point in a 2D scenario (Figure 3).

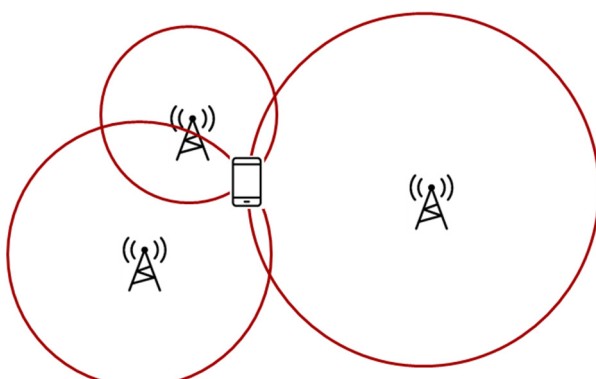

**Figure 3.** The method of determining the location of the transmitter by means of trilateration.

There are several implementation solutions in the ToF system infrastructure, active and passive. A popular configuration is the active implementation, i.e., Master-Slave, where the Master device controls communication and sends packets, and the Slave responds after a strictly defined time. In this way, the control device is able to calculate the delay in arrival of the response packet. The time is counted between the transmission of the control packet and the response received from the slave device, then the value of the fixed and predetermined fixed travel time is subtracted.

The accuracy of the determining the location depends on the number of the calculations in the procedure in a given time interval. The higher the frequency that ToF measurements result in, the greater the precision (Figure 4).

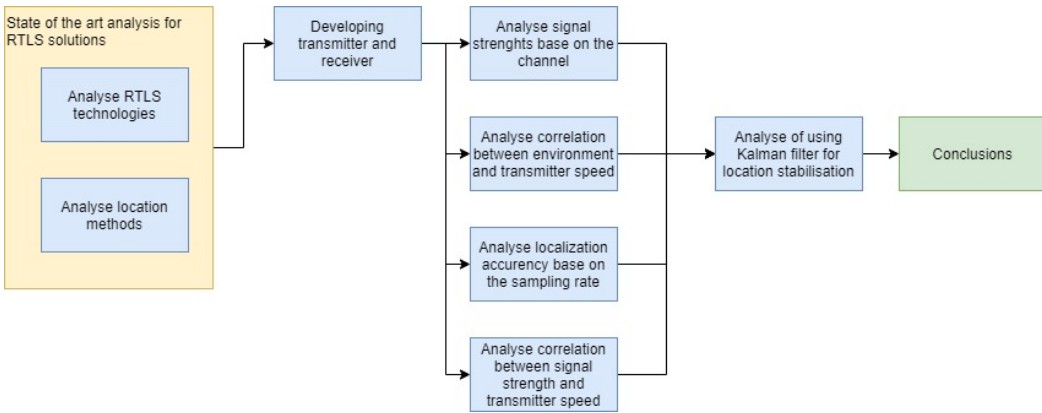

**Figure 4.** Research methodology.

*3.3. Materials*

3.3.1. Receiver

The main task of the receiver is to scan continuously to pre-process raw BLE packets and send them for further analysis via Ethernet using one of the communication protocols. For this reason, an important component in this element of the system is the BLE module and the radio system.

When selecting the electronic part of the receiver, many variants of solutions were considered and tested.

The ESP32 system was selected to test the first solution. The core of the presented board is the ESP-WROOM-32 module. The integrated circuit has universal I/O pins that can be configured in the form of interfaces such as: I2C, SPI and UART. ESP32 is adapted for communication in the 2.4 [GHz] band and in the Bluetooth BLE standard version v4.2. The board is powered via a standard microUSB 5 [V] connector. The great advantage of the module is its easy availability and low price. The technical parameters of the device are presented in the Table 2 below.

**Table 2.** Technical parameters of the receiver system based on the ESP32 module.

| Parameter | Value |
|---|---|
| Supply voltage | 5 [V] |
| SRAM memory | 520 [KB] |
| Flash memory | 16 [MB] |
| Network security | WEP, WPA/WPA2, PSK/Enterprise, AES/SHA2/Elliptical Curve Cryptography/RSA-4096 |
| Low-level interfaces | 3xUART, 3xSPI, 2xI2C (2xI2S), 12-channel ADC converter, 2-channel DAC converter, PWM outputs, SD card interface |
| Additional | Hall sensor, Touch interface |

During the tests, devices from several companies that met the design assumptions were considered. The Table 3 below shows a comparison of 4 test chips meeting the initial requirements for the receiver.

**Table 3.** Summary of the main tested receiver modules.

| Name | Raspberry Pi | Raspberry Pi | Nano Pi | Banana Pi |
|---|---|---|---|---|
| Model | 3 Model B | 3 Model B + | M1 Plus | M2 + |
| Processor | Broadcom BCM2837 64-bit | Broadcom BCM2837B0 64bit | Allwinner H3 | Allwinner H3 |
| Core | Quad-Core ARM Cortex A53 | Quad-Core ARM Cortex A53 | Quad-Core ARM Cortex A7 | Quad-Core ARM Cortex A7 |
| Operating system | - Linux Raspbian, - Windows 10 IoT | - Linux Raspbian, - Windows 10 IoT | - u-boot, - Debian, - Ubuntu Mate, - Ubuntu Core | - Ubuntu, - Raspbian |
| CPU clock | 1.2 [GHz] | 1.4 [GHz] | 1.2 [GHz] | 1.2 [GHz] |
| Architecture | ARMv8-A | ARMv8-A | ARMv7-A | ARMv7-A |
| RAM | 1 GB LPDDR2 @ 900 MHz | 1 GB LPDDR2 @ 900 MHz | 1 GB | 1 GB DDR3 |
| Memory | MicroSD cartridges | MicroSD cartridges | - microSD card, - 8 GB eMMC | - microSD card, - 8 GB eMMC |
| Power | 5.1 [V]/2.5 [A] | 5.1 [V]/2.5 [A] | 5 [V]/2 [A] | 5 [V]/2 [A] |
| Power input | microUSB | - microUSB, - 5 [V] GPIO, - PoE with an additional overlay | microUSB | DC 4.0/1.7 [mm] socket |
| | 85 × 56 × 17 [mm] | 85 × 56 × 17 [mm] | 65 × 65 [mm] | 65 × 65 [mm] |
| Network interface | - Ethernet 10/100 [Mbps] port, - 802.11 b/g/n 150 [Mbps] | - Ethernet 10/100 [Mbps] port, - 802.11 b/g/n 150 [Mbps] | - Ethernet 10/100/1000 [Mbps] port, - 802.11 b/g/n | - Ethernet 10/100/1000 [Mbps] port - 802.11 b/g/n, |
| Bluetooth | BLE v4.1 | BLE v4.2 | BLE 4.0, dual mode | BLE 4.0 |
| USB | 4x USB 2.0—socket type A | 4x USB 2.0—socket type A | 3x USB 2.0—one in GPIO | 2x USB 2.0 |
| Communication | UART, SPI, I2C, GPIO | UART, SPI, I2C, GPIO | UART, SPI, I2C, GPIO, PWM | UART, I2C, SPI |

RaspberryPi enabled the quickest and most hassle-free pre-testing procedure. Compared to the rest of the devices, it has a more refined and matched system, as well as the greatest support. Based on the RaspberryPi and the Bluetooth module in the form of *LaunchPad CC2640R2* connected in series, a special receiver was prepared, which was named the *Localisation Test Box*.

The prepared device was a prototype allowing us to more precisely define the requirements for the receiver, resulting from practical tests. In addition, it initially allowed us to both exclude and familiarize with the proposed solutions regarding the location method, as well as all the elements related to it. The photo of the *Localization Test Box* is shown below (Figure 5).

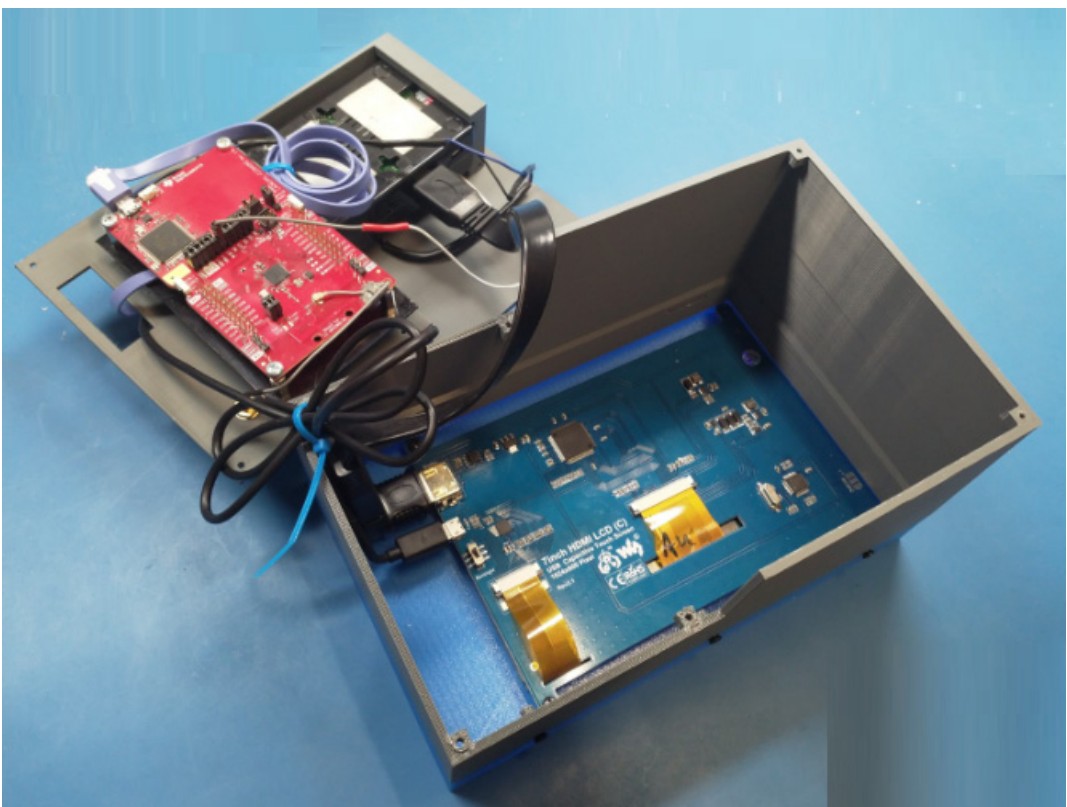

**Figure 5.** Photo of Localization Test Box.

### 3.3.2. Transmitter

The transmitter is one of the key elements of the system, because this module will be assigned to the tracked object and then registered by the receivers. After a thorough review of ready-made solutions available on the market, just like in the case of a receiving device, it was decided to make our own electronics. Our own design of the circuit board made it possible to be versatile, which in the event of required changes or problems will allow for a wider range of modifications and expansion of the device. During research and testing, several versions of the device were made based on the first guidelines.

The transmitter will also be called SKK Hive Beacon or SKK Hive Sensor, with sensors. Each of the iterations presented is based on the SaBLE-x-R2 module. The developed transmitter—SKK Hive Beacon—according to our assumptions, is a miniature wireless radio device operating in the Bluetooth Low Energy (BLE) standard in version v4.0, v4.1, v4.2, and in the future v5.0. The devices made have EEPROM memory, which allows one to save data from sensors, the purpose of which is to increase the accuracy of the location tracked using the accelerometer and pressure gauge. The power range of the device depends on the radio power settings and the antenna used. The device software includes many functions that facilitate the analysis and determination of the device location, such as:

- Marking of broadcasting channels
- Changing the frequency of sending basic BLE (Advertisement Interval) packages
- Configurable BLE parameter settings
- Ability to change broadcasting channels
- AoA (Angle of Arrival) Mode
- ToF (Time of Flight) mode

The heart of the device is the LSR Sable-X chip, which includes the CC2640R2 microprocessor with the power supply and antenna system.

CC2640R2 is a wireless device suitable for Bluetooth Low Energy applications. MCU is a member of the CC26xx family, which is characterized by low energy consumption, which ensures long device operation time using a simple button battery.

CC2640R2 consists of two processors. The main ARM Cortex-M3 (48 MHz) is responsible for controlling peripherals, the radio system and the Transmitter Controller. The second ARM Cortex-M0 microcontroller is responsible for controlling the 2.4 GHz radio.

The transmitter module has a LIS2DH accelerometer made with Ultra Low-Power technology, which measures acceleration in three axes. The device has configurable interrupts by two independent events, such as fall and motion detection. It also has the ability to detect 4D/6D. The device is available in a small LGA plastic case.

The accelerometer used can work in many modes thanks to the use of the FIFO queue. In the sensor, the Stream Mode with support for the interrupt routine is activated. This allows you to save energy for the main system—CC2640R2. The main processor only downloads data when an interruption occurs, i.e., when a given acceleration value is exceeded. It is also possible to continuously record the acceleration in each of the three axes and the resultant one.

Device parameters:

- Voltage range: 1.71–3.6 [V]
- Current consumption on the level: 2 [μA]
- Measuring range: ±2 g, ±4 g, ±8 g and ±16 g
- Interfaces: I2C, SPI
- Reading frequency: 1 [Hz]–5.3 [kHz]

## 4. Research and Results

### 4.1. Analysis of the Influence of Broadcasting Channels on Signal Quality

The analysis focused on the influence the channels to the signal quality. The tests were performed with the use of transmitters with embedded software (00: 12: 6F: C2: 1C: 2E, 00: 12: 6F: C2: 1C: 2F, 00: 12: 6F: C2: 1C: 2C). Embedded software enables marking of broadcasting channels. The measurement distance was 5 m and the measurement time was 25 min. In the test, the distance was 5 m and the measurement time was 30 min.

The number of readings on individual channels at a distance of 5 m is shown in Figures 6–8.

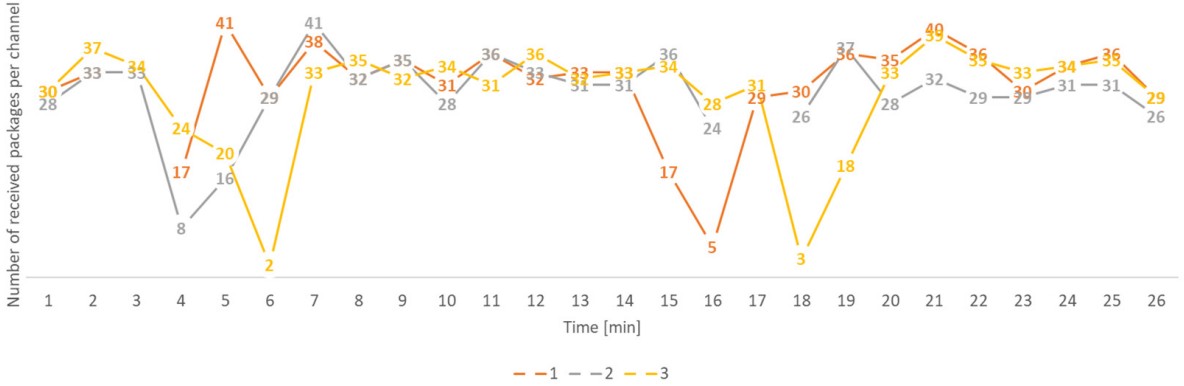

**Figure 6.** Number of readings depending on the transmit channel for the transmitter 00: 12: 6F: C2: 1C: 2C, distance 5 m.

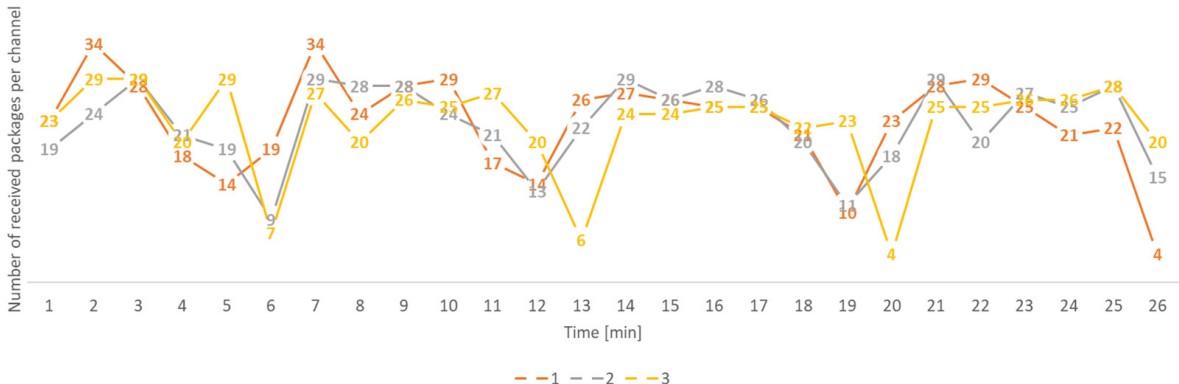

**Figure 7.** Number of readings depending on the transmission channel for the transmitter 00: 12: 6F: C2: 1C: 2E, distance 5 m.

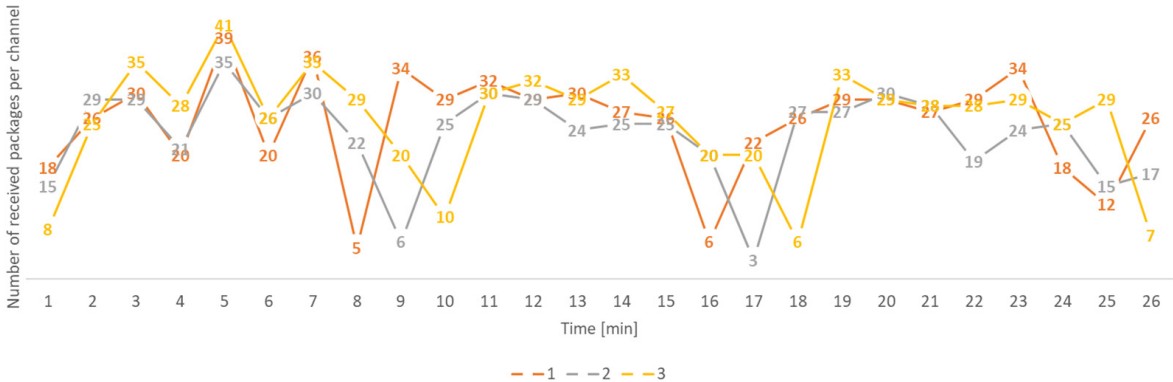

**Figure 8.** Number of readings depending on the transmission channel for the transmitter 00: 12: 6F: C2: 1C: 2F, distance 5 m.

From the Figures 6–8, there is no significant difference between the number of readings on a given transmit channel for individual transmitters. Instability in the number of readings may result from momentary disturbances on a given channel or problems on the receiver side.

The previously noticeable cyclicality of the decrease in the number of measurements, occurring every 7–9 min, did not occur during the tests at night. This precludes a malfunction of the receiver. The cyclicality was caused by the influence of the environment—a Wi-Fi hotspot which was disturbed by radio waves on the BLE frequency.

*4.2. Correlation of the Environment with the Speed of Movement of Objects*

The analysis of the surroundings with the speed of movement of objects was performed for an object moving with the speed of walking in an open square. The speed of the object was about 3.6 km/h. The object moved along a curve defined by points numbered 1 to 57 as shown in the drawing (Figure 9).

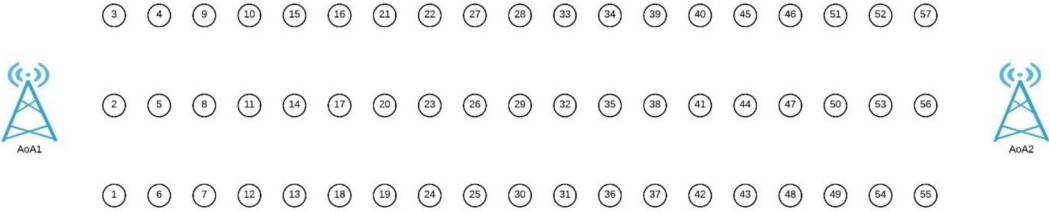

**Figure 9.** Arrangement of measurement points in relation to AoA receivers.

For the presented diagram, the signal reception angle from the transmitter and the signal level were measured. The angle measurement error in degrees is shown below (Figure 10).

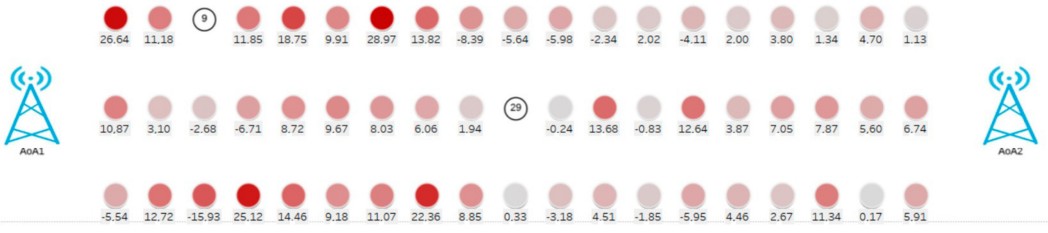

**Figure 10.** Error in measuring the angle to the AoA receiver, measured in degrees.

It is noticeable that the greatest measurement error occurs at a distance of up to 12 m (Figure 11). Above this distance, the measurement angle error most often falls within 5°, which is a satisfactory result. The accuracy of the measurement is twice as high with the use of a second receiver. The measurement results are presented below (Figure 12).

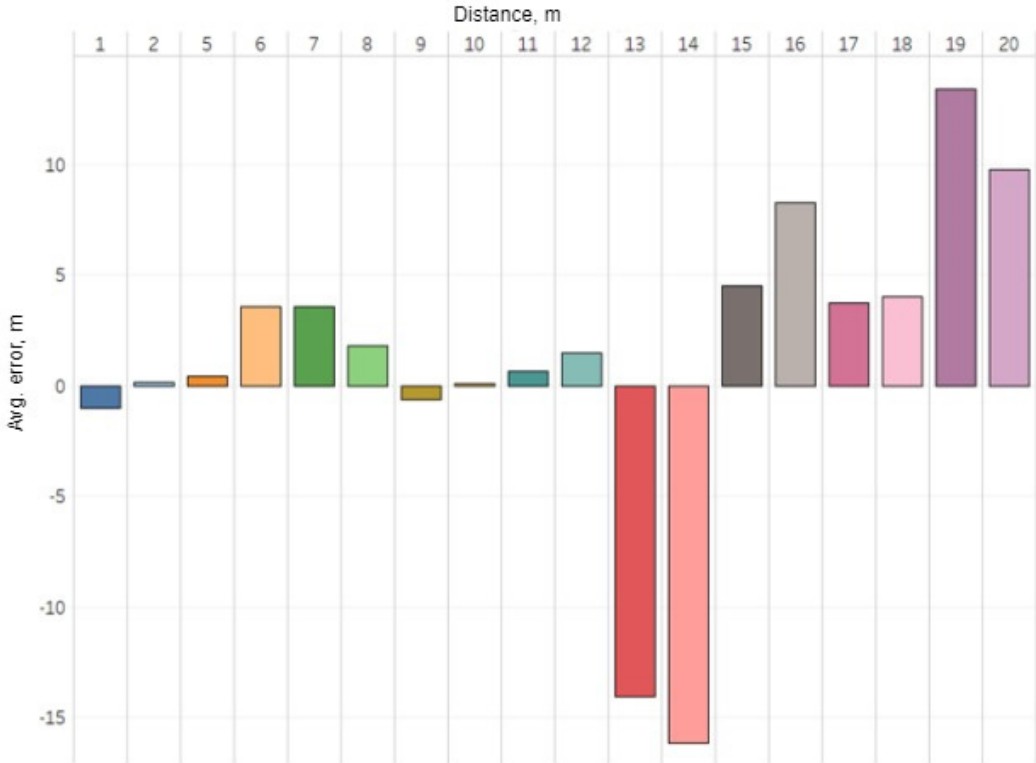

**Figure 11.** Measurement angle error in meters at individual measurement points.

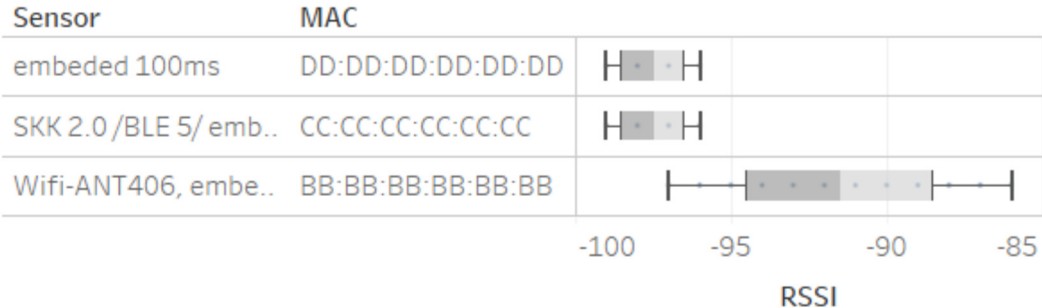

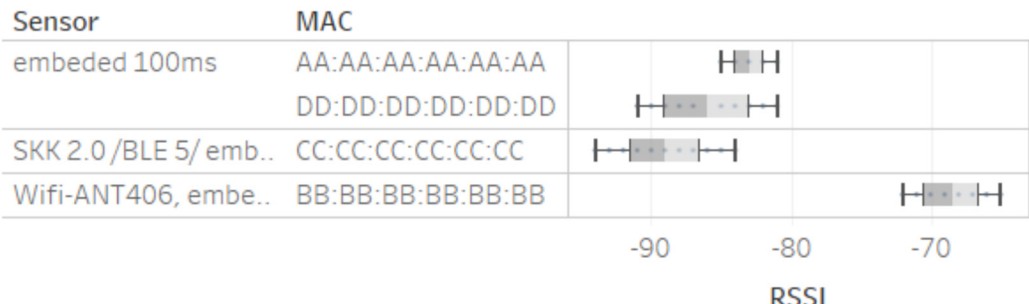

**Figure 12.** The mustache box diagram for the comparison of sensors (measurement from two receivers at different distances: top figure, distance 40 m, receiver No. 2; bottom figure, distance 10 m, receiver No. 1).

Based on the collected data, it can be concluded that the application of smoothing the results by means of the lower and upper quartile cut-offs will allow to achieve an acceptable measurement error. The dependence of the RSSI signal level on the distance can be used to stabilize the signal level and obtain a more precise location of objects.

### 4.3. The Sampling Rate Required to Achieve the Specified Level of Localization Accuracy

The sampling rate was tested for two BLE modules: Raspberry and ESP32.

The average values of the RSSI signal level for the entire time interval from 1 to 10 s (Table 4), an increase in the average signal level decrease for:

- Raspberry (approx.):
  a.  1.3 dBm, 2.1% between 1 s and 5 s,
  b.  1.4 dBm, 2.2% between 1 s and 10 s,
- ESP32 (approx.):
  a.  2.4 dBm, 3.8% between 1 s and 5 s,
  b.  2.7 dBm, 4.3% between 1 s and 10 s.

On the other hand, the minimum values have been increased for:

- Raspberry (approx.):
  a.  7 dBm, 8.9% between 1 s and 5 s,
  b.  8.9 dBm, 11.3% between 1 s and 10 s,
- ESP32 (approx.):
  a.  11 dBm, 12.1% between 1 s and 5 s,
  b.  14 dBm, 15.4% between 1 s and 10 s.

**Table 4.** Dependence of the RSSI level on the length of the scan time, dBm.

| Receiver | Value | Time of Scanning | | | | | | | |
|---|---|---|---|---|---|---|---|---|---|
| | | **1 s** | **2 s** | **3 s** | **4 s** | **5 s** | **6 s** | **8 s** | **10 s** |
| Raspberry | Mean | −63.4 | −62.1 | −62.3 | −62.1 | −62.1 | −62.0 | −62.0 | −62.0 |
| | Min | −78.6 | −74.8 | −73.8 | −72.5 | −71.6 | −71.9 | −71.8 | −69.7 |
| | Max | −49.0 | −49.0 | −49.0 | −49.0 | −49.0 | −49.0 | −49.0 | −49.0 |
| ESP32 | Mean | −62.3 | −60.6 | −60.4 | −60.1 | −59.9 | −59.8 | −59.7 | −59.6 |
| | Min | −91.0 | −88.0 | −80.0 | −80.0 | −80.0 | −76.0 | −76.5 | −77.0 |
| | Max | −52.0 | −52.0 | −52.0 | −52.0 | −52.0 | −52.0 | −52.0 | −52.0 |

Aggregating the results from the time interval (5–10 s) to the averaged RSSI value additionally allows one to improve the signal quality by 4–15% depending on the length of the time interval in which the aggregation was performed.

During the measurements performed with the settings of the transmission frequency every 200 ms, it was possible to receive 2.8 packets per second. The remaining packages have not been received. Thus, a maximum of three signal levels per second can be obtained. It is a value sufficient to achieve the assumed goals.

The measurement of the number of transmitters detected by the receiver in a specific scanning time interval was performed for the Raspberry receiver and the ESP32 system (Table 5). The number of BLE transmitters in the room was 8. Measurements were made at a distance of 5 m.

**Table 5.** Number of scanned transmitters in a specific time period.

| Receiver | Value | Scan Time | | | | | | | |
|---|---|---|---|---|---|---|---|---|---|
| | | **1 s** | **2 s** | **3 s** | **4 s** | **5 s** | **6 s** | **8 s** | **10 s** |
| Raspberry | Mean | 6.7 | 7.6 | 7.7 | 7.8 | 7.8 | 7.8 | 7.8 | 7.9 |
| | Min | 2 | 5 | 5 | 6 | 6 | 6 | 6 | 7 |
| | Max | 8 | 8 | 8 | 8 | 8 | 8 | 8 | 8 |
| ESP32 | Mean | 1.9 | 2.6 | 3.1 | 3.6 | 3.6 | 3.7 | 3.8 | 3.9 |
| | Min | 1 | 1 | 1 | 2 | 2 | 2 | 2 | 3 |
| | Max | 4 | 4 | 4 | 4 | 4 | 4 | 4 | 4 |

General observations for the analysed systems:

- ESP32-has a slow scan speed
- ESP32- has frequent, unpredictable jamming of the system. Reason unknown.
- ESP32-receives transmitters with a signal about 2 dBm smaller than Raspberry.

The influence of the signal level on the frequency of scanning transmitters in the room was tested using the Raspberry RnD receiver and the ESP32 devkit system. The number of BLE transmitters in the room was eight. Measurements were made at a distance of 3 m, 5 m and 8 m. The results are presented in Table 6.

**Table 6.** Dependence of the number of scanned devices on the RSSI level.

| | | | Distance | | | | | | | | | | | |
| --- | --- | --- | --- | --- | --- | --- | --- | --- | --- | --- | --- | --- | --- | --- |
| | | | 3 m | | | | 5 m | | | | 8 m | | | |
| | | | Scan Time | | | | | | | | | | | |
| | | | 1 s | 3 s | 5 s | 10 s | 1 s | 3 s | 5 s | 10 s | 1 s | 3 s | 5 s | 10 s |
| Raspberry | Number of transm | Mean | 6.6 | 7.6 | 7.7 | 7.8 | 6.6 | 6.9 | 7.0 | 7.0 | 6.3 | 7.5 | 7.6 | 7.7 |
| | | Min | 1 | 5 | 6 | 6 | 1 | 5 | 6 | 6 | 1 | 5 | 6 | 6 |
| | | Max | 8 | 8 | 8 | 8 | 8 | 8 | 8 | 8 | 8 | 8 | 8 | 8 |
| | RSSI level | Mean | −62.3 | −62.4 | −62.4 | −62.4 | −63.4 | −62.3 | −62.1 | −62.0 | −68.1 | −68.2 | −68.2 | −68.2 |
| | | Min | −77.6 | −68.8 | −67.0 | −65.4 | −78.6 | −73.8 | −71.6 | −69.7 | −79.7 | −74.6 | −74.6 | −72.3 |
| | | Max | −52.0 | −57.4 | −59.5 | −59.7 | −49.0 | −49.0 | −49.0 | −49.0 | −49.0 | −60.0 | −62.8 | −64.6 |
| | Number of transmi | Mean | 1.9 | 3.1 | 3.6 | 3.8 | 1.6 | 1.9 | 2.1 | 3.6 | 1.6 | 2.5 | 3.1 | 3.5 |
| | | Min | 1 | 1 | 1 | 2 | 1 | 1 | 2 | 3 | 1 | 1 | 2 | 3 |
| | | Max | 4 | 4 | 4 | 4 | 4 | 4 | 4 | 4 | 4 | 4 | 4 | 4 |
| ESP32 devkit | RSSI level | Mean | −64.4 | −64.4 | −64.3 | −64.4 | −62.3 | −60.4 | −59.9 | −59.6 | −66.3 | −66.4 | −66.4 | −66.3 |
| | | Min | −86.0 | −80.3 | −71.6 | −67.9 | −91.0 | −80.0 | −80.0 | −77.0 | −95.0 | −90.0 | −86.0 | −84.6 |
| | | Max | −53.0 | −53.0 | −55.7 | −59.7 | −52.0 | −52.0 | −52.0 | −52.0 | −53.0 | −53.0 | −54.3 | −55.2 |

*4.4. Correlation of Other Moving Objects with the Speed of the Analysed Objects*

Signal fluctuations in radio technologies are a major technological problem. The key is the proper selection of transmission and reception parameters, the selection of antennae and the method of signal processing that allows for its stabilization. To select the parameters of the devices, the sampling frequency should be considered depending on the type of object. The higher the speed of the object, the smaller the sampling should be. However, at a certain sampling rate, it is not possible to improve the signal quality, so constantly increasing the sampling rate is not a good solution. In order to select the appropriate sampling frequency and Bluetooth parameters, several field measurements were made, examples of which are shown below.

In order to determine the required sampling frequency, an algorithm was prepared that changes the Bluetooth parameters, such as: Maximum Scan Responses, Scan Interval, Scan Time, Scan Window (Table 7). 180 s was set as the measurement time on one setting. Several measurements were made. The set of settings with the best parameters is shown in the table below.

**Table 7.** The analyzed Bluetooth parameter settings.

| Number of Research Analysis | Duration of One Iteration | Maximum Scan Responses | Scan Interval | Scan Time | Scan Window |
|---|---|---|---|---|---|
| 1 | 180 | 20 | 48 | 100 | 32 |
| 2 | 180 | 20 | 96 | 100 | 64 |
| 3 | 180 | 20 | 144 | 100 | 96 |
| 4 | 180 | 20 | 192 | 100 | 128 |
| 5 | 180 | 20 | 240 | 100 | 160 |

The measurement was made for the prepared transmitter prototypes with different settings and antennae. The research used, among others: the BLE 5.0 standard; a modified version of the software (embedded) with transmission settings every 100 ms and 200 ms; sensor version 2.0/BLE 5.0; sensors with ceramic antennae; and the Larid SaBLE-x-R2 module and external Winizen type antennae, ANT406. The measurement results for selected antennae and sensors are presented below (Figure 5).

The best result was obtained for the sensor with the antenna with the ANT406 antenna, but for the receiver located at a distance of 40 m, an increased range was observed in which the signal level fluctuated (−97 to −86 db), which can be limited by cutting the lower and upper quartiles (−88 to −94 db), i.e., the signal level varies in the range of ±6 db. For comparison, the sensors with the 100 ms version of the embedded software and the sensors transmitting the signal in the BLE 5.0 standard had a signal range within ±2 dB, but were much weaker, by 18 dB at the shorter distance and 6 dB at the longer distance.

By analysing the parameters of Bluetooth, itself, tt can be concluded that the best results were obtained for setting no. 5, i.e.,:

Maximum Scan Responses: 20

- Scan Interval: 240
- Scan Time: 100
- Scan Window: 160

The highest number of packets were received for such settings (they were rarely lost) (Tables 8 and 9).

**Table 8.** Number of packets received for receiver No. 2.

| File Name (Parameters) | Count of RSSI | Std. Dev. of RSSI |
|---|---|---|
| 3 | 1046 | 0.774782666 |
| 1 | 1171 | 3.611268972 |
| 2 | 1342 | 3.410815032 |
| 4 | 1402 | 0.864016223 |
| 5 | 1542 | 0.734495244 |

**Table 9.** Number of packets received for receiver No. 1.

| File Name (Parameters) | Count of RSSI | Std. Dev. of RSSI |
|---|---|---|
| 3 | 992 | 7.814493358 |
| 1 | 3861 | 8.218871628 |
| 4 | 4335 | 7.857758797 |
| 2 | 4600 | 8.848465938 |
| 5 | 6170 | 7.959578686 |

The quality of the received signal is shown in the figure below (Figure 13) in the form of a mustache frame graph.

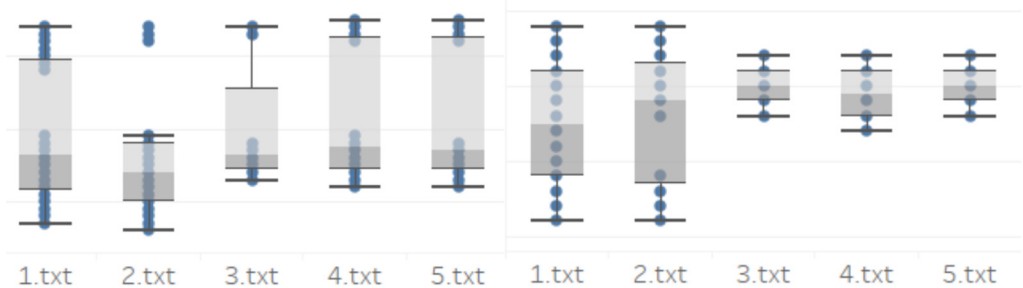

**Figure 13.** Signal quality for receivers (receiver No. 2 on the right, receiver No. 1 on the left).

As can be seen for receiver No. 2 located at the greatest distance, the signal quality assessed due to its fluctuations was the best for parameters No. 3, 4 and 5. However, for receiver No. 1, the quality was the best for parameters No. 1, 4, 5. However, taking into account the number of packets received, it should be clearly stated that the signal quality is the best for parameter No. 5.

The signal quality is best for these Bluetooth settings:

- Maximum Scan Responses: 20
- Scan Interval: 240
- Scan Time: 100
- Scan Window: 160

Directional antennae were also used to develop the prototype of the system. The dependence of the angle divergent from the real angle as a function of distance was determined.

From the Table 10 it can be seen that the acceptable measurement angle error is up to 5°. For such an error, it is still possible to achieve measurement accuracy of up to 3 m. These calculations do not consider signal stabilization, which will increase the accuracy of the location.

**Table 10.** Dependence of the location accuracy on the measurement angle error and distance from the receiver (without angle stabilization).

|  | $\alpha\,[°]$ | Distance of the Measuring Line from the Receiver [m] | Distance of the Transmitter from the Receiver [m] | Measurement Error [m] |
|---|---|---|---|---|
| Actual angle | ±30 | 10 | 5.8 | - |
|  | ±5 | 10 | 7.0 | 1.2 |
| Discrepancy | ±10 | 10 | 8.4 | 2.6 |
|  | ±20 | 10 | 11.9 | 6.1 |
| Actual angle | ±30 | 15 | 8.7 | - |
|  | ±5 | 15 | 10.5 | 1.8 |
| Discrepancy | ±10 | 15 | 12.6 | 3.9 |
|  | ±20 | 15 | 17.9 | 9.2 |
| Actual angle | ±30 | 24 | 13.9 | - |
|  | ±5 | 24 | 16.8 | 2.9 |
| Discrepancy | ±10 | 24 | 20.1 | 6.3 |
|  | ±10 | 24 | 20.1 | 6.3 |
| Actual angle | ±30 | 40 | 23.1 | - |
|  | ±5 | 40 | 28.0 | 4.9 |
| Discrepancy | ±10 | 40 | 33.6 | 10.5 |
|  | ±20 | 40 | 47.7 | 24.6 |

Further research related to the number of nearby objects and their mutual interference will be carried out at the stage of implementing the system pilot.

### 4.5. Kalman Filter and Its Influence on Distance Estimation

Signal measurements in the target work environment were carried out in a printing materials warehouse; a photograph of them is shown in Figure 14. Four transmitters were used for it, which moved along a designated track and stopped at a distance of 1 m, 3 m, 5 m, 10 m, 15 m, 20 m from the receiver. The idle time was 5 min, and the transmission interval was 1 s. As a result, for each sensor, more than 300 RSSI signal strength measurements were obtained for each distance. The measured signal level is shown in the Figure 14 below.

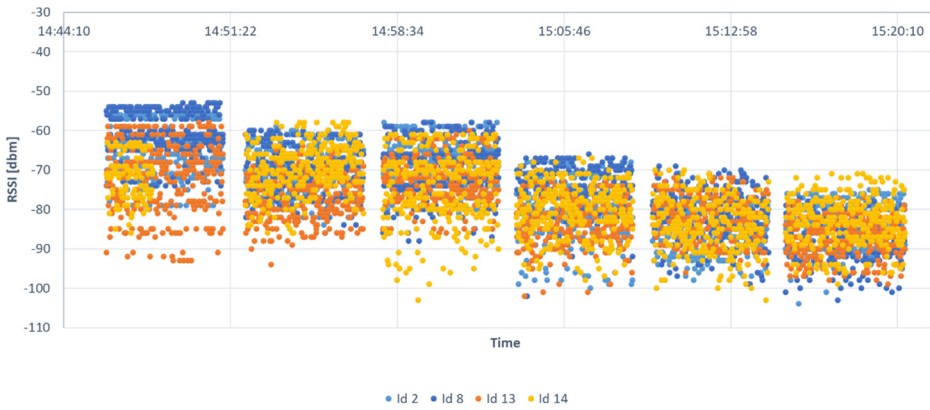

**Figure 14.** Signal strength (RSSI) at a distance of 1 m, 3 m, 5 m, 10 m, 15 m, 20 m from the receiver.

A Kalman filter was used to stabilize the signal level. The diagram showing its operation is presented Figure 15 below.

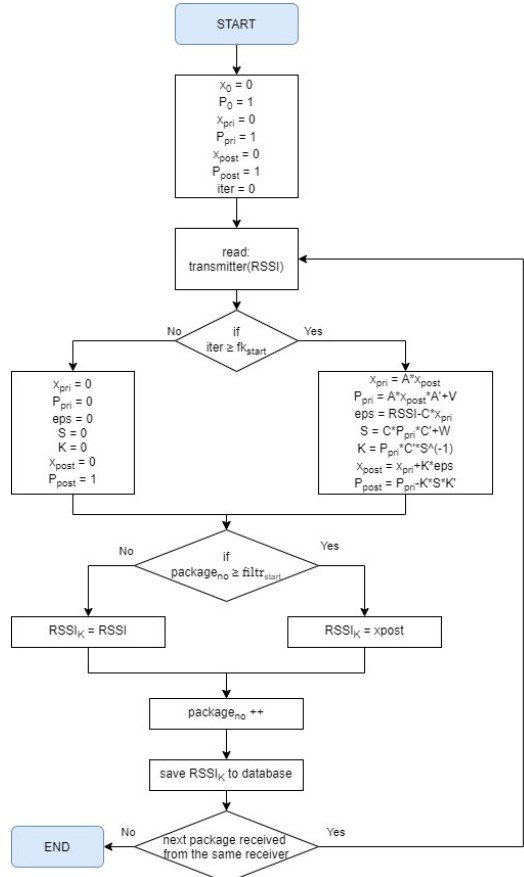

**Figure 15.** Diagram of Kalman filter.

where:

$t_{pack}$—time interval between packets, assumed value = 0.1

$fk_{start}$—number of the packet from which the filter starts, assumed value = 10

A [1,1] = 1-state matrix

C [1,1] = 1-output matrix

$st_{dev}$-standard deviation, assumed value = 10

v RSSI noise correction factor, assumed value = 30

V = v * st_dev * t_pack-RSSI noise

w = st_dev ˆ 2-measurement noise

The matrices of state, input and output were taken as the value of 1. As the values of the sensors are known.

The noise covariance matrix plays a very important role in the case of the Kalman filter. There is a relationship between the measurement noise and the Kalman gain. The greater the measurement noise, the smaller the Kalman gain value. The Kalman gain plays an important role in the final result, because in case of a large noise value, the Kalman gain will be smaller, so the new RSSI value should not be much different from that determined in the first phase.

The measurement noise is calculated by the square of the standard deviation value. Process noise is calculated by the product of the smoothing value, standard deviation and the measurement frequency.

The figures below show the graphs for the entire measurement period for all distances with the smoothing constant amounting to a maximum of 200 and a minimum of 10. According to the assumption that for a large value of the noise covariance matrix, the Kalman gain will be greater, for the smoothing constant equal to 200, the Kalman gain is approximately 0.954, while for the smoothing constant of 10, the Kalman gain was about 0.620. The orange colour shows the signal after applying the Kalman filter, while the blue colour shows the measured signal without using the Kalman filter.

The diagram (Figure 16) below shows that the measured signal almost coincides with the signal in which the Kalman filter was applied, but the signal fluctuations are much smaller. You can on average change the position of the sensor every 300 measurements, so there are significant limits when the position is changed. The farther away the point, the greater the value of the RSSI signal, so the signal is less well received.

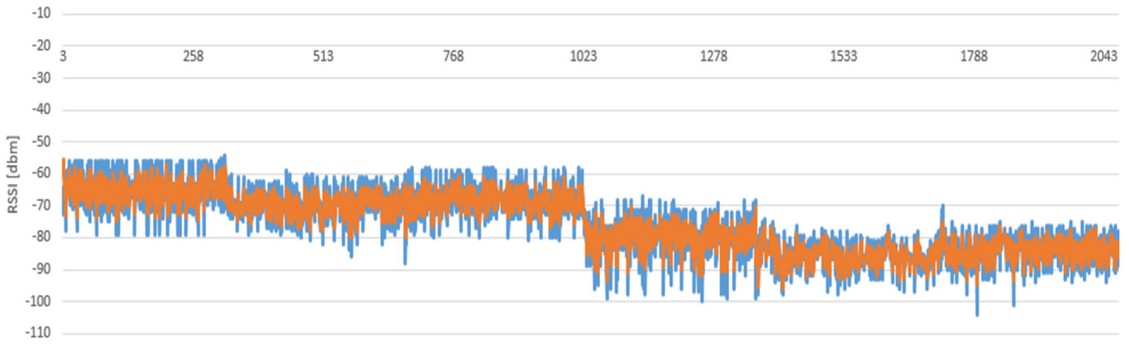

**Figure 16.** RSSI for raw data (blue) and after using Kalman filter (orange).

All RSSI mean values for individual noise, as well as for the measured value for the sensors, are presented in the tabular form below (Table 11).

**Table 11.** RSSI value for the sensors.

| Distance [m] | 200 | 100 | 50 | 20 | 10 |
|---|---|---|---|---|---|
| 1 | −65.04 | −65.03 | −65.01 | −64.96 | −64.88 |
| 3 | −70.12 | −70.12 | −70.12 | −70.13 | −70.13 |
| 5 | −68.21 | −68.21 | −68.21 | −68.22 | −68.22 |
| 10 | −80.03 | −80.03 | −80.03 | −80.02 | −80.02 |
| 15 | −86.10 | −86.10 | −86.09 | −86.09 | −86.08 |
| 20 | −83.78 | −83.79 | −83.79 | −83.79 | −83.79 |
| Total: | −75.67 | −75.67 | −75.67 | −75.66 | −75.64 |

The values in the table above also change to the hundredth level, which can affect the distance calculation. However, it can be seen that the mean value for the measured RSSI is almost identical to that after running the Kalman filter.

Comparing the above graph with the graphs for the same distance obtained in the analysis of the data of other transmitters, it is noticeable that the signal strength reaches −90 dbm, so it is close to the receiver's range limit.

Based on the results obtained for the Kalman filter, the distance of the transmitters from the receiver and the measurement errors were calculated. The data of all measured smoothing constants for the sensors were used for this. The calculations are presented in the tabular form below (Table 12). The determination of measurement errors for the smoothing constant value 200 is shown below (Table 13).

**Table 12.** Measurement error without applying Kalman filter.

| | RSSI | | | | | | |
|---|---|---|---|---|---|---|---|
| d [m] | 1 | 3 | 5 | 10 | 15 | 20 | Mean: |
| d [m] counted | x | 2.6 | 3.8 | 9.8 | 17.5 | 21.6 | 11.0 |
| measurement error [m] | x | 0.5 | 1.2 | 0.2 | 2.4 | 1.6 | 1.2 |
| measurement error [%] | x | 15.0 | 24.2 | 1.7 | 16.3 | 8.0 | 13.0 |
| measurement error, max [m] | | | | 2.4 | | | |
| RSSI | −65.0 | −70.1 | −68.2 | −80.0 | −86.1 | −83.8 | −75.5 |
| $n=$ | x | 1.1 | 0.5 | 1.5 | 1.8 | 1.4 | 1.2 |

**Table 13.** Measurement error after applying Kalman filter.

| | RSSI | | | | | | |
|---|---|---|---|---|---|---|---|
| d [m] | 1 | 3 | 5 | 10 | 15 | 20 | Mean: |
| d [m] counted | x | 2.80 | 4.30 | 9.70 | 16.50 | 21.40 | 10.94 |
| measurement error [m] | x | 0.20 | 0.70 | 0.30 | 1.50 | 1.40 | 0.82 |
| measurement error [%] | x | 7.14 | 16.28 | 3.09 | 9.09 | 6.54 | 8.43 |
| measurement error, max [m] | | | | 1.5 | | | |
| RSSI | −63 | −69.2 | −71.2 | −78.8 | −85.2 | −82.6 | −77.40 |
| $n=$ | x | 1.1 | 0.5 | 1.5 | 1.8 | 1.4 | 1.26 |

## 5. Discussion

The signal strength fluctuation for the receiver is between 7 and 10 dBm for ESP32 device and between 13 and 14 dBm for Raspberry. For commercial implementation, the number of devices scanned in the time window is also important. For Raspberry, the optimal time window is 5 s; in this time six transmitters can be detected. ESP32 has problem with detecting devices in short time, as just two transmitters can be detected in 4–8 s time window. The localisation precision depends on the distance between transmitter and receiver, and the angle from axis of the directional antenna. For the distance of 10 m the measurement error is 1.2–6.1 m, whilst for the distance of 40 m, the measurement error is 4.9 to 24.6 m. Using the Kalman filter can reduce the localization error to 1.5 m.

Location systems based on BLE, RFID or Wi-Fi technologies have a measurement accuracy of about 3–5 m. However, this accuracy depends on the environment in which they work; the more metal and water around the transmitters, the greater the risk of signal disturbance. The accuracy of the location is influenced by other devices in the vicinity, the direction of the antenna, the signal strength, the speed of moving objects as well as the technical condition of the device itself.

A faulty device can easily change the accuracy of the calculated AoA information by interfering with the packet structure. This is because devices with a Bluetooth version less than 5.0 do not impose any security requirements. An RTLS with the Angle of Arrival makes location tracking easy with greater precision. BLE is ideal for zonal or room-level tracking.

## 6. Conclusions

The solution presented in the paper is based on our own developed transmitter. The standard model is inaccurate and cannot be implemented. Presented research confirms that the location system based on the Bluetooth technology can be implement in industry. The location based on the Bluetooth is hard to predict. Signal disruption impact is caused by other devices, steal, containers with water and more. However, proper data processing can stabilize the RSSI impact on the location precision. The maximum measurement error when using data pre-processing and the Kalman filter is about 1.5 m. The presented research was used to develop IT system. This system has been working for few months in the private company.

As part of the development of work and further analysis, it is planned to implement position sensors (magnetometer, accelerometer and gyroscope). In addition, devices should be based on the newest BLE version (5.4).

The limitation of the current solution is the latency of the determined position in relation to the real time. Additionally, there is relatively high energy consumption, which can be reduced by implementing energy saving mechanisms.

**Author Contributions:** Conceptualization: A.L. and M.C.; methodology, A.L.; software, M.C.; validation, A.L., J.S. and M.C.; formal analysis, A.L.; investigation, A.L., J.S. and M.C.; resources, J.S.; data curation, A.L.; writing—original draft preparation, A.L.; writing—review and editing, M.C. and J.S.; visualization, A.L.; supervision, A.L.; project administration, J.S. and A.L.; funding acquisition, A.L., J.S. and M.C. All authors have read and agreed to the published version of the manuscript.

**Funding:** This research was funded by project RPMP.01.02.01-12-0497/16, "SKK Hive Analysis-System for proactive management and analysis of resources".

**Institutional Review Board Statement:** Not applicable.

**Informed Consent Statement:** Informed consent was obtained from all subjects involved in the study.

**Data Availability Statement:** Data is a business secret.

**Acknowledgments:** The presented research is carried out as part of the project: RPMP.01.02.01-12-0361/19 "Smart Logistics Unit for IoT and Industry 4.0-a logistic unit using artificial intelligence to optimize processes".

**Conflicts of Interest:** The authors declare no conflict of interest.

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
