# Peer review of "Real-Time Location System (RTLS) Based on the Bluetooth Technology for Internal Logistics"

_sustainability, doi:10.3390/su15064976_

Round 1
Reviewer 1 Report
MDPI Sustainability Journal (Manuscript ID: sustainability-2170384)
Comments to the Author
This paper proposes a real-time location system using Bluetooth technology for internal logistics. It is an important topic however, there are several points need to be addressed to improve the quality of the manuscript.
Suggestions to improve the quality of the paper are provided below:
1) Please provide the full address of the authors’ affiliations. (Line 5-6).
Comments for the abstract:
2) Current version of the abstract needs to be restructured and significantly improved. As such:
a. The abstract has been split into two paragraphs which is not a common practice in scientific papers. Please combine these two paragraphs into one.
b. The problem statement is very disjoint and there is no proper flow of the motivation, methodology and results. It really needs to be rewritten to follow a proper order.
c. R&D (Line 16) and IT (Line 19) need to be defined before using the acronyms.
d. There is no need to mention the specific company name (SKK S.A.) in the abstract, instead, please mention it as a private company.
e. Please include numerical results of the findings in the abstract.
3) There are quite a lot of grammatical errors and typos in the manuscript. Please ensure that the manuscript has been proofread by a native English speaker before submitting it. Some examples:
· For the title and rest of the article, please correct “Real time” “Real-time”
· All of the acronyms need to be explained before using them. For example Line 29, RFID, UWB, Line 41 BLE and others. Please make sure this has been corrected for the rest of the manuscript.
· Line 26, “GPS / Glonass / Beidou” “GPS/Glonass/Beidou”
· Line 46, “km / h” “km/h”
· Line 40, “Lan / Wi-Fi” “Lan/Wi-Fi”.
· Line 54, merge these 3 references as [1,2,3].
· Line 60, clinic’s’ -> “clinics’”
· Line 101, extra space between “on the”.
· Line 107, “pac-kets” ->”packets”.
· Line 133, “Localistation” “Localisation”
4) Although this paper uses Bluetooth technology for a chosen technology, the related work (state of the art) section does not really contain a comprehensive background for BLE and needs to be significantly improved. Please include a paragraph explaining:
- BLE technology definition and specifics
- Advantages and disadvantages of BLE
- Related and popular applications using BLE in buildings such as occupancy detection, smart grid, smart energy management, and evacuation measurements. Please refer to the suggested established works as below:
BLE for occupancy detection
Tekler, Z.D., Low, R., Gunay, B., Andersen, R.K. and Blessing, L., 2020. A scalable Bluetooth Low Energy approach to identify occupancy patterns and profiles in office spaces. Building and Environment, 171, p.106681.
BLE used for smart grid applications
Collotta, M. and Pau, G., 2015. A novel energy management approach for smart homes using bluetooth low energy. IEEE Journal on selected areas in communications, 33(12), pp.2988-2996.
BLE for smart energy management
Tekler, Z.D. et al., 2022. Plug-Mate: An IoT-based occupancy-driven plug load management system in smart buildings. Building and Environment, 223, p.109472.
BLE for evacuation measurements
Astarita, V., Festa, D.C., Giofrè, V.P. and Vitale, A., 2018. Bluetooth portal‐based system to measure the performance of building emergency evacuation plans and drills. IET Intelligent Transport Systems, 12(4), pp.294-300.
5) In the Introduction section, I strongly suggest that the authors to include the contributions in bullet points to clearly highlight the novelty of the paper and how each contribution improves upon the existing literature.
6) Certain figures are redundant by themselves. Such as Figure 2 and Figure 3. If authors would like to visualize the current setup, I suggest merging these two and indicating several components that would be worth indicating. On top of that, please do not use the word “photo” for the description of Figure 3. Instead, use the word “figure”.
7) Table 6 does not fit into the page margin and floats. It should be arranged properly. Similarly, first column of Table 12, needs to be readjusted to fit in.
8) Conclusion section needs to be significantly improved by including the future directions, limitations of the existing approach, and implications of the proposed solution.
9) For the author contributions, please remove “For research articles with several authors, a short paragraph specifying their individual contributions must be provided. The following statements should be used” and only directly state the contributions by starting from “Conceptualization, A. Lorenc etc. …”
Author Response
Dear reviewer,
Thank you for your suggestions to improve the quality of the paper. We made following changes:
1) Please provide the full address of the authors’ affiliations. (Line 5-6).
We add the full address of the authors’ affiliations.
Comments for the abstract:
2) Current version of the abstract needs to be restructured and significantly improved. As such:
- The abstract has been split into two paragraphs which is not a common practice in scientific papers. Please combine these two paragraphs into one.
We rewrite the abstract. We combine two paragraphs into one.
- The problem statement is very disjoint and there is no proper flow of the motivation, methodology and results. It really needs to be rewritten to follow a proper order.
We changed the structure of the paper and added the diagram showing research methodology.
- R&D (Line 16) and IT (Line 19) need to be defined before using the acronyms.
We add the definition for the acronyms.
- There is no need to mention the specific company name (SKK S.A.) in the abstract, instead, please mention it as a private company.
We change it in the text.
- Please include numerical results of the findings in the abstract.
We add the numerical results in of the findings from research part.
3) There are quite a lot of grammatical errors and typos in the manuscript. Please ensure that the manuscript has been proofread by a native English speaker before submitting it. Some examples:
- For the title and rest of the article, please correct “Real time” à“Real-time”
- All of the acronyms need to be explained before using them. For example Line 29, RFID, UWB, Line 41 BLE and others. Please make sure this has been corrected for the rest of the manuscript.
- Line 26, “GPS / Glonass / Beidou” à“GPS/Glonass/Beidou”
- Line 46, “km / h” à“km/h”
- Line 40, “Lan / Wi-Fi” à“Lan/Wi-Fi”.
- Line 54, merge these 3 references as [1,2,3].
- Line 60, clinic’s’ -> “clinics’”
- Line 101, extra space between “on the”.
- Line 107, “pac-kets” ->”packets”.
- Line 133, “Localistation” à“Localisation”
We correct the typos and grammatical errors listed above and similar ones in the paper.
4) Although this paper uses Bluetooth technology for a chosen technology, the related work (state of the art) section does not really contain a comprehensive background for BLE and needs to be significantly improved. Please include a paragraph explaining:
- BLE technology definition and specifics
We add the following specifications:
The Bluetooth Low Energy (BLE) is a wireless technology designed and marketed by the Bluetooth Special Interest Group (Bluetooth SIG). BLE is designed for devices with low power consumption that operates in the 2.4 GHz ISM (The Industrial, Scientific and Medical) band. The goal of this technology is to connect devices over relatively short range. It is currently a very popular protocol widely used in IoT (Internet of Things), Industry 4.0 and Smart Home. In addition, BLE technology support localisation solution such as AoA (Angle of Arrival), AoD (Angle of departure).
And we extend the Bleutooth Technology specification in table 1.
- Advantages and disadvantages of BLE
We add the following information of BLE:
The main advantages of BLE can be: very low power consumption, security - encryption using 128 bit AES (Advanced Encrypotion Standard) algorithms, cheap to implement, up to 255 bytes per package (message capacity), frequency hopping partially eliminates interference from other networks. Main disadvantages are lower data rates compared to Wifi or GSM (Global System for Mobile Communications) technology, cannot be use for long distance and vulnerable to interception.
- Related and popular applications using BLE in buildings such as occupancy detection, smart grid, smart energy management, and evacuation measurements. Please refer to the suggested established works as below:
BLE for occupancy detection
Tekler, Z.D., Low, R., Gunay, B., Andersen, R.K. and Blessing, L., 2020. A scalable Bluetooth Low Energy approach to identify occupancy patterns and profiles in office spaces. Building and Environment, 171, p.106681.
BLE used for smart grid applications
Collotta, M. and Pau, G., 2015. A novel energy management approach for smart homes using bluetooth low energy. IEEE Journal on selected areas in communications, 33(12), pp.2988-2996.
BLE for smart energy management
Tekler, Z.D. et al., 2022. Plug-Mate: An IoT-based occupancy-driven plug load management system in smart buildings. Building and Environment, 223, p.109472.
BLE for evacuation measurements
Astarita, V., Festa, D.C., Giofrè, V.P. and Vitale, A., 2018. Bluetooth portal‐based system to measure the performance of building emergency evacuation plans and drills. IET Intelligent Transport Systems, 12(4), pp.294-300.
We add the suggested papers.
5) In the Introduction section, I strongly suggest that the authors to include the contributions in bullet points to clearly highlight the novelty of the paper and how each contribution improves upon the existing literature.
We add the following information:
In the paper the unique research was made to analyse the disruptions and prob-lems for the using BLE technology for RTLS. This research include:
- Analyse signal strengths base on the channel,
- Analyse correlation between environment and transmitter speed,
- Analyse localization accurency base on the sampling rate,
- Analyse correlation between signal strength and transmitter speed.
Also the developing works was made for production of electronic devices: trans-mitter and receiver.
6) Certain figures are redundant by themselves. Such as Figure 2 and Figure 3. If authors would like to visualize the current setup, I suggest merging these two and indicating several components that would be worth indicating. On top of that, please do not use the word “photo” for the description of Figure 3. Instead, use the word “figure”.
We deleted the figure 2 and 3.
7) Table 6 does not fit into the page margin and floats. It should be arranged properly. Similarly, first column of Table 12, needs to be readjusted to fit in.
We adjust table 12 and 13 but table 6 is to wide for adjusting.
8) Conclusion section needs to be significantly improved by including the future directions, limitations of the existing approach, and implications of the proposed solution.
We add the following information:
As part of the development of work and further analysis, it is planned to implement position sensors (magnetometer, accelerometer and gyroscope). I addition, devices should be based on the newest BLE version 5.4.
The limitation of the current solution is the latency of the determined position in relation to the real time. Additionally, there is relatively high energy consumption, which can be reduced by implementing energy saving mechanisms.
9) For the author contributions, please remove “For research articles with several authors, a short paragraph specifying their individual contributions must be provided. The following statements should be used” and only directly state the contributions by starting from “Conceptualization, A. Lorenc etc. …”
We delete unnecessary text.
Reviewer 2 Report
The topic of the article Real time location system (RTLS) based on the Bluetooth technology for internal logistics is relevant and interesting. The authors have a good idea, but there are some mistakes in the article.
Details to be corrected:
1 1) You need to organize the abstract, write it coherently.
2) Many abbreviations are used but not explained.
3) Keywords - also need to be rewritten, because abbreviations cannot be included in them either.
4) The introduction (lines 26-27) mentions GPS, Glonass, Beidou technology, but does not mention Galileo.
5) It is not clear what the purpose of the article is. It should be added in the introduction.
6) The introduction cannot contain any subsection (1.1.). This sub-section should be supplemented and renamed as the scientific literature analysis section or similar.
7) At the end of the introduction, the structure of the article (arrangement of chapters and subsections) must be presented.
8) In line 284, the numbering of the figures is incorrect.
9) Figure 11-13 are blurry, especially Figure 13. Maybe it can be discarded.
10) There is no discussion section, but it is necessary.
11) The Conclusion section is very short.
12) References are incorrect and not in accordance with the requirements. More literature sources from prestigious journals such as Symmetry, Sustainability and others need to be added.
13) The name is written in lower case in position 10 of the list.
14) At the end of the article, I missed further research on this topic.
Author Response
Dear reviewer,
Thank you for your suggestions to improve the quality of the paper. We made following changes:
1) You need to organize the abstract, write it coherently.
We rewrite the abstract.
2) Many abbreviations are used but not explained.
We add the explained the abbreviations.
3) Keywords - also need to be rewritten, because abbreviations cannot be included in them either.
We changed the abbreviation to the full words.
4) The introduction (lines 26-27) mentions GPS, Glonass, Beidou technology, but does not mention Galileo.
We add the information about Galileo.
5) It is not clear what the purpose of the article is. It should be added in the introduction.
We add the information of unique research presented in the paper.
6) The introduction cannot contain any subsection (1.1.). This sub-section should be supplemented and renamed as the scientific literature analysis section or similar.
We changed the structure of the paper.
7) At the end of the introduction, the structure of the article (arrangement of chapters and subsections) must be presented.
We describe the structure of the article at the end of introduction
8) In line 284, the numbering of the figures is incorrect.
We update the figures numbering.
9) Figure 11-13 are blurry, especially Figure 13. Maybe it can be discarded.
We changed the figure quality.
10) There is no discussion section, but it is necessary.
Nadal nie mamy skecji discussion, dajemy końcówkę conclusion and discussion?
We add discussion section.
11) The Conclusion section is very short.
We extend the conclusion works and add the future research.
12) References are incorrect and not in accordance with the requirements. More literature sources from prestigious journals such as Symmetry, Sustainability and others need to be added.
We add new references
13) The name is written in lower case in position 10 of the list.
We correct it
14) At the end of the article, I missed further research on this topic.
We extend the conclusion works and add the future research.
Round 2
Reviewer 1 Report
Thank you for addressing my comments and concerns. I have noticed a minor reference formatting issue on Reference [2]. Compared to others, it is missing the complete author list. Please correct it based on the information below and ensure all the references follow the same format.
[2] Tekler, Z.D., Low, R., Yuen, C. and Blessing, L., 2022. Plug-Mate: An IoT-based occupancy-driven plug load management system in smart buildings. Building and Environment, 223, p.109472.
After addressing this issue, the manuscript will be ready for publication. Great job!
Author Response
Dear reviewer,
Thank you for your work and suggestions. We correct the literature.
Reviewer 2 Report
The authors have revised almost everything in the article based on my comments and suggestions. There are still some inaccuracies left.
Please look at lines 315-319.
Sonclusions and discussion sections should also be expanded. Limitation should be moved to the discussion section.
Author Response
Dear reviewer,
Thank you for your work and suggestions.
We rewrite the paragraph at lines 315-319 and expend the discussion section.